# EgoMem: Lifelong Memory Agent for Full-duplex Omnimodal Models

## Abstract

We introduce EgoMem, the first lifelong memory agent tailored for full-duplex models that process real-time omnimodal streams. EgoMem enables real-time models to recognize different users from raw audiovisual streams, to provide personalized response, and to maintain long-term knowledge of users' facts, preferences, and social relationships extracted from audiovisual history. EgoMem operates with three asynchronous processes: (i) a *retrieval* process that dynamically identifies user via face and voice, and gathers relevant context from a long-term memory; (ii) an *omnimodal dialog* process that generates personalized audio responses based on the retrieved context; and (iii) a *memory management* process that automatically detects dialog boundaries from omnimodal streams, and extracts necessary information to update the long-term memory. Unlike existing memory agents for LLMs, EgoMem relies entirely on raw audiovisual streams, making it especially suitable for lifelong, real-time, and embodied scenarios. Experimental results demonstrate that EgoMem's retrieval and memory management modules achieve over 95% accuracy on the test set. When integrated with a fine-tuned RoboEgo omnimodal chatbot, the system achieves fact-consistency scores above 87% in real-time personalized dialogs, establishing a strong baseline for future research.

## 1 Introduction

A wide range of AI applications involve lifelong omnimodal streams. A notable example is a robot deployed in homes and public spaces (AgiBot-World-Contributors et al., 2025; Bu et al., 2025). In similar scenarios, the models are required not only to follow instructions swiftly, but also to recognize users, remember their histories, understand social relationships, and deliver personalized services. Technically, the crucial capabilities to meet these requirements include *omnimodality*, *real-time responsiveness*, and *humanoid cognition* (Wang & Sun, 2025). For *real-time responsiveness*, there have been solutions to achieve full duplexity, either based on time-division multiplexing (Wang et al., 2024; Zhang et al., 2024b), or on native duplex (Défossez et al., 2024; Yao et al., 2025a) schemes. Yet, *humanoid cognition* remains an underexplored capability for current omnimodal, full-duplex systems. In this work, we study the lifelong memory capability as a critical step towards *humanoid cognition*, since memory is the foundation of both human and advanced artificial intelligence (Jimenez Gutierrez et al., 2024). We focus on real-time personalized dialog as a major task to validate the effectiveness of lifelong memory in omnimmodal scenarios.

We showcase the role of lifelong memory in personalized omnimodal dialogs as follows. (1) When a user Emily shows up, a polling process detects the user identity as Emily directly from the audiovisual stream (*e.g.,* camera and microphone inputs); (2) The profile of Emily is encoded and put into the dialog context of an omnimodal chatbot; (3) When Emily asks "does any of my colleagues love tennis?", a query regarding the relation "colleague" and keyword "tennis" is generated by the chatbot, activating a textual retrieval to the knowledge base containing Emily's social relation graph, which returns a dialog record of "John, colleague, 2024-05-13, user discussed a tennis game he played 2 days ago". This record is further encoded as dialog context; (4) The chatbot answers "Yes, Emily, your colleague John loves tennis" based on the available context; (5) The system extracts user facts: "Emily shows interest in tennis", and dialog record "2024-05-14, user asked if any of her colleagues loves tennis.", from the raw audiovisual stream of the recent dialog, and updates Emily's profile memory with these contents for future use; (6) When Emily shows up on another day, the model is able to greet with: "Hi Emily, did you talk to John about tennis?".

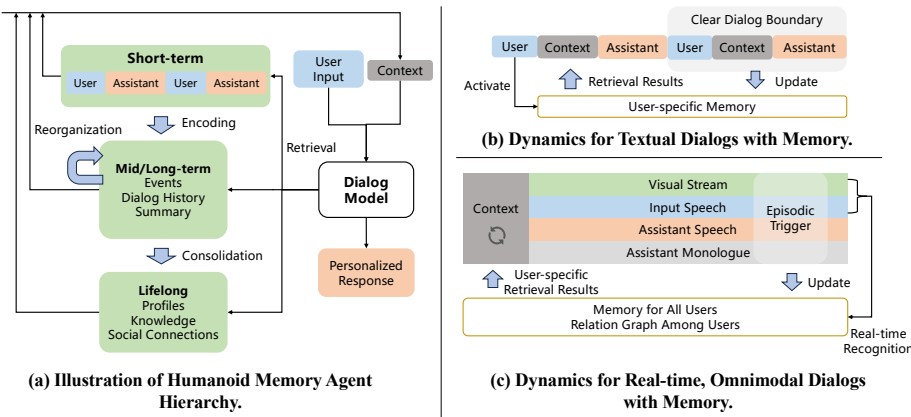

Figure 1: **Textual memory agents vs. full-duplex omnimodal memory agents (ours).**

In literature, two primary approaches have been explored to equip textual large language models (LLMs) with long-term memory: extended context windows (Su et al., 2021; Wu et al., 2024) and memory agents (Zhong et al., 2024; Chhikara et al., 2025; Xu et al., 2025b; Kang et al., 2025). However, neither method transfers well to *lifelong omnimodal* scenarios. On one hand, extended context windows can retain long sequences encoding full omnimodal information (He et al., 2024). Yet, in lifelong settings, the length of audiovisual streams grows without bound, making even million-token contexts insufficient (Men et al., 2024). On the other hand, memory agent methods (Figure 1 (a)) are well-suited for lifelong operation (Lee et al., 2024; Wang & Chen, 2025; Li et al., 2025b), but typically rely on several strong assumptions: user identities are explicitly known, dialog sessions have clear boundaries, and all inputs are textual (Figure 1 (b)). Unfortunately, these assumptions do not hold in full-duplex omnimodal applications (Défossez et al., 2024; Zhang et al., 2024a; Lin et al., 2025), in which the user identities are implicitly encoded in audiovisual streams, and there is no well-defined boundaries for dialog turns or user sessions (Figure 1 (c)). Furthermore, existing memory agents generally overlook the multiuser social relation graph (Au et al., 2025), which is an important element for humanoid cognition in lifelong scenarios.

To address these issues, we propose EgoMem, the first lifelong memory system tailored for omnimodal scenarios and designed to facilitate full-duplex personalized dialog. EgoMem operates through three asynchronous processes. First, a **Retrieval Process** is responsible for real-time polling recognition of users, implemented with an audiovisual retrieval mechanism. It also contains a content-driven text retrieval module to gather related textual documents. This process facilitates efficient integration of both user-specific and RAG-style (Gao et al., 2023) information into the dialog flow. Second, an **Omnimodal Dialog Process** uses a fine-tuned dialog model to deliver full-duplex, personalized responses in real time, grounded in the retrieved context. Third, a **Memory Management Process** handles dialog boundary detection, information extraction, and memory updating, based on real-time raw audiovisual streams. This process ensures up-to-date memory over time.

We integrate EgoMem memory system to RoboEgo (Yao et al., 2025b), a *native* full-duplex model that is best aligned with our target scenarios. We fine-tune RoboEgo under our EgoMem framework to deliver real-time, lifelong, and personalized responses to arbitrary user. We conduct automated evaluations across the audio, textual, and visual retrieval modules, the memory management module, and the system's personalization abilities, considering either single-user profile (Level-1) and multi-user social graph (Level-2) as reference contexts. Evaluation results show that these modules exhibit high accuracy and robustness, and that the incorporation of EgoMem enhances personalization without compromising RoboEgo's original dialog capabilities.

Our contributions are as follows: (1) *framework*: we propose EgoMem, a lifelong memory agent for full-duplex, omnimodal interaction, which to the best of our knowledge is the first of its kind; (2) *implementation*: we provide a concrete implementation of EgoMem based on the RoboEgo backbone, including detailed module designs, data construction pipelines, and training configurations; (3) *evaluation*: we demonstrate that EgoMem achieves robust performance on personalization tasks in lifelong omnimodal scenarios, establishing a solid baseline for future research.

## 2 PRELIMINARIES

### 2.1 FULL-DUPLEX OMNIMODAL MODELS

Full-duplex omnimodal models are able to process real-time audiovisual inputs and demonstrate capabilities to listen and speak simultaneously. In each time step $t$, a full-duplex omnimodal model $F$ takes as input a listening audio $a_t$, video frames $v_t$, and optionally a textual input $l_t$. It generates a slice of spoken audio response $r_t$:

$$r_t = F_\theta(a_t, v_t, l_t). \tag{1}$$

We adopt RoboEgo (Yao et al., 2025b) as our primary dialog model, as it supports a *native* full-duplex scheme at least for audio. The *native* scheme features lower response latency and better scalability (Défossez et al., 2024; Lin et al., 2025; Yao et al., 2025a), compared to time-division multiplexing (TDM) schemes. Also, in general instruction-following tasks, RoboEgo's response quality and user experiences are comparable to state-of-the-art systems such as Qwen-2.5-Omni (Xu et al., 2025a).

In RoboEgo, both the listening and speaking stream are processed with a frame rate of 12.5 fps, each frame corresponding to one autoregressive forward step $t$. In each step, 17 tokens are merged into one embedding: the listening and speaking audio frame are both encoded by 8 tokens, and the text channel contributes 1 token. Please refer to Appendix A for more details on the model's structure and stream organization. Note that EgoMem's framework and methodology can be applied to other full-duplex omnimodal models $F$ or to different organizations of $a_t$, $v_t$, and $l_t$ beyond our implementation.

### 2.2 MEMORY AGENT PARADIGM

EgoMem is designed to facilitate full-duplex, personalized chat for lifelong-deployed omnimodal models. As a first step, we focus on the case where there is only one active speaker at a time, leaving the more complex cocktail party problem (Haykin & Chen, 2005) for future work. In this setting, in each time step $t$, the main dialog model $F$ takes two additional inputs: the user profile $p_t$, and reference information $c_t$:

$$r_t = F_\theta(a_t, v_t, l_t, p_t, c_t). \tag{2}$$

Here, $p_t$ and $c_t$ are encoded in the text channel, commonly referred to as the *context* or *short-term memory* in current literature, delivered to $F$ as part of its KV-cache (Vaswani et al., 2017; Dao, 2023).

EgoMem manages a textual memory $M$ with three core functions: *retrieval*, *writing*, and *updating*.

*Retrieval.* The retrieval function provides $p_t$ and $c_t$ to the dialog model by searching related information in $M$ based on current dialog content:

$$p_t, c_t = \text{EgoMem.retr}(a_t, v_t, M). \tag{3}$$

In traditional textual memory agents (Gao et al., 2023), $p_t$ is naively accessible from user accounts, while the retrieval process for $c_t$ is always activated after the user's textual input. In lifelong omnimodal scenarios, however, both user identities and dialog boundaries are implicit. The memory agent should directly detect user identities and session boundaries from raw audiovisual streams.

*Writing.* The writing function extracts important events from lifelong multimodal streams and stores them in the memory unit $M$:

$$Episode = \text{EgoMem.extract}(a_{0\sim t}, v_{0\sim t}, l_{0\sim t}), \tag{4}$$

$$M \leftarrow \text{EgoMem.write}(M, Episode). \tag{5}$$

In EgoMem, memory writing is asynchronous to the main dialog, executed through independent processes. Unlike traditional memory agents, EgoMem takes as input the raw multimodal stream fragments in the dialog history ("episodic memory"), and extracts textual descriptions for the events, user persona, and other useful information.

*Updating.* EgoMem periodically performs online or offline memory consolidation: it integrates existing memory into new, consolidated representations, and solves potential conflicts:

$$M \leftarrow \text{EgoMem.update}(M). \tag{6}$$

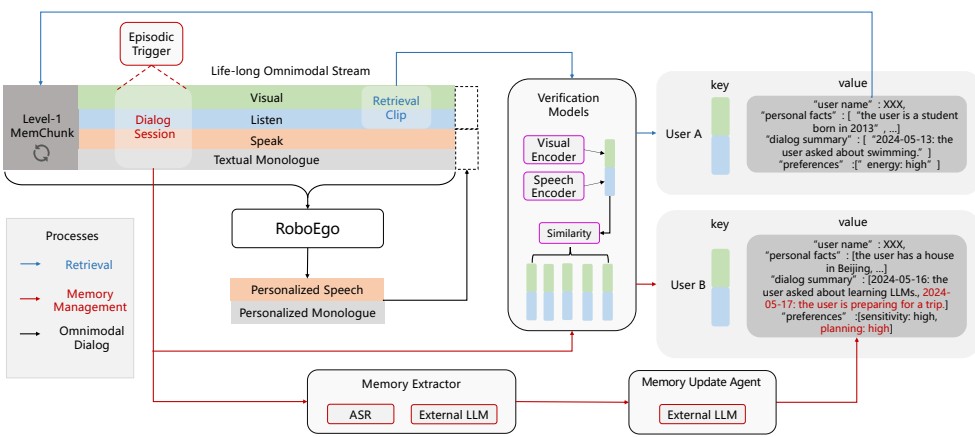

**Figure 2: System illustration for EgoMem Level-1 (Profile-only).**

## 3 EGOMEM

The design of EgoMem can be divided to two levels: Level-1 EgoMem facilitates profile-based personalization; and Level-2 EgoMem supports additional references such as users' social networks. Compared to Level-1, Level-2 is particularly suited for application scenarios where users are more interconnected, such as home robots. We introduce the implementation of each level based on our dialog flow outlined in Section 2.1.

### 3.1 LEVEL-1 EGOMEM: PROFILE-ONLY

In Level-1, $M$ contains only the profile information for each of its recognized users. Formally, it sets $c_t$ = None in equations 2 and 3. The profile of each user in a key-value pair: the *key* contains one visual embedding $v_f^u$ for face verification and one audio embedding $v_s^u$ for speaker verification; the *value* is a dictionary storing the users' name, personal facts, summary of previous dialogs, and preferences. The operation flow of Level-1 EgoMem is illustrated in Figure 2. It is driven by the three component processes running asynchronously: the *retrieval* process (Section 3.1.1), the *omnimodal dialog* process (Section 3.1.2), and the *memory management* process (Section 3.1.3).

### 3.1.1 RETRIEVAL

To identify the current user, a retrieval process runs in a lifelong "polling" manner with fixed intervals of 2 seconds. This is a critical design enabling the dialog model to *actively start* talking to the user (e.g., "*Greetings John!*"). In every 2 seconds, EgoMem processes a chunk of audio and visual signals with length $\tau$, and extracts query vectors:

$$v_f^q = \text{visual\_encoder}(v_{(t-\tau):t}), \tag{7}$$

$$v_s^q = \text{speech\_encoder}(a_{(t-\tau):t}). \tag{8}$$

$v_f^q$ and $v_s^q$ are used for face and speaker verification with each *user*'s key ($v_f^u$ and $v_s^u$), respectively. If a valid user is found, its profile is tokenized with a textual tokenizer and pushed to the textual channel of a special *Level-1 MemChunk* field in the main dialog's token stream (Figure 2, top left). This special chunk occupies a maximum length of 512 time steps, with its 16 audio tokens always filled with *<empty>*. It is attended by every forward pass of the RoboEgo model. Note that *Level-1 MemChunk* is managed exclusively by the retrieval process: its textual content is refreshed only when the current recognized user differs from the previous one. Every time *Level-1 MemChunk* is refreshed, EgoMem triggers one additional forward pass for RoboEgo that updates the KV cache for the entire dialog history. This operation introduces ignorable overhead in inference.

**Face Verification.** We leverage an open-sourced pipeline from DeepFace (Serengil & Ozpinar, 2024) to extract faces from video frames. Specifically, we use Retinaface (Deng et al., 2020) as

a face detection backend, and Facenet512 (Schroff et al., 2015) as the visual encoder, resulting in 512-dimensional face features. The retrieval quits if no face is detected; otherwise, we first find the closest existing user $u$ with the minimal cosine distance $d = 1 - cosine\_similarity(v_f^q, v_f^u)$ to the query vector $v_f^q$, and then verify with a pre-tuned threshold $\delta = 0.3$:

$$\text{current user} = \begin{cases} u, & \text{if } d < \delta, \\ \text{new user}, & \text{else.} \end{cases} \tag{9}$$

**Speaker Verification.** We leverage a wavelm_large (Chen et al., 2022) model fine-tuned specifically for speaker verification (Anastassiou et al., 2024) as our speech feature extractor, producing 256-dimensional $v_s^q$ and $v_s^u$ vectors. We combine cosine similarity with adaptive s-norm (Karam et al., 2011; Cumani et al., 2011) for best performance and robustness (Section 5.1).

### 3.1.2 OMNIMODAL DIALOG

This is the main process running the RoboEgo chat service. *Level-1 MemChunk* is attended by RoboEgo in each step. If no user profile is returned by the retrieval process, *Level-1 MemChunk*'s text channel is filled with *<pad>* tokens. We fine-tune RoboEgo with the corresponding streaming data format (Section 4) to generate personalized spoken responses based on the user's profile information.

### 3.1.3 MEMORY MANAGEMENT

The *memory management* process instantiates EgoMem's extract, write, and update functions (eq. 4 - 6). With a fixed time interval, it conducts content extraction on the 17-way audio-language token stream from the main dialog process. For a 8192-step stream chunk ($\sim$11 minutes) in history, each time step is labeled by a sequence tagging model (namely **Episodic Trigger**) to mark the boundaries of dialog sessions for each user. Next, an external LLM, serving as **Memory Extractor**, is prompted to extract events, user facts, and user preferences from the fragmented streams of each session. Afterwards, the memory management process calls the retrieval functions to identify the user of this session. If a *new* user is found, EgoMem creates a new memory item in $M$, stores the face/speech embedding as keys, and initializes the user's profile with the extracted contents. Otherwise, the user identity and the extracted memory contents are provided to a **Memory Update Agent**, which is an external LLM prompted to figure out potential conflicts and update the user's profile in $M$.

**Episodic Trigger.** The episodic trigger is used to find the boundaries of dialog sessions in which the user's identity is consistent. It not only detects the start and end of dialog sessions, but also splits the sessions from different users. Specifically, the episodic trigger predicts tags for each time step in the audio stream (the aligned listen and speak audio tokens):

$$\text{Tag}_{0 \sim t} = \text{episodic\_trigger}(a_{0 \sim t}, r_{0 \sim t}). \tag{10}$$

Specifically, the episodic trigger assigns a label to each time step with the following paradigm: {0: *no dialog*; 1: *start of a new user's dialog session*; 2: *in-session step*; 3: *end of current user's session*}. The detailed model structure is explained in Appendix B.1.

The **Memory Extractor** and **Memory Update Agent** are also detailed in Appendix B.1.

### 3.2 LEVEL-2 EGOMEM: CONTENT-DRIVEN

In Level-2 EgoMem, $M$ maintains not only the user profiles, but also the social relation graph among them. For each user, we add a field containing a list of triplets representing the graph edges from the current user to others. Optionally, any other useful information can be added to $M$ for a similar RAG processes. Formally, Level-2 EgoMem provides both $p_t$ and $c_t$ in equations 2 and 3. While $p_t$ comes from a *polling* user recognition, $c_t$ comes from the the primary model (RoboEgo)'s *active* retrieval to $M$. We exemplify Level-2 EgoMem in Figure 3, showing its major differences to Level-1. We specify the comparison to Level-1 for each of the core processes (Section 3.1.1 - 3.1.3) as follows:

**Level-2 Retrieval.** As shown in Figure 3, the *Level-1 MemChunk* maintains its function in the Level-2 system; it is driven by the external polling retrieval process. An *Level-2 MemChunk* with

**Figure 3: System illustration for EgoMem Level-2 (Content-driven).** We focus on showing the differences in retrieval process and hide the details for other processes like memory management.

a maximum length of 256 is added to the reserved field in the stream; unlike *Level-1 MemChunk*, *Level-2 MemChunk* is driven by active textual queries from the RoboEgo dialog model. For example, in a dialog session between user A and RoboEgo, *Level-1 MemChunk* is kept the same (user A's profile). In contrast, after each user instruction, RoboEgo can activate an independent **Textual Retrieval** process to memory $M$. A textual query is generated by RoboEgo on its monologue channel based on the current dialog. The retrieval result is tokenized and cached in *Level-2 MemChunk*. The implementation of this **Textual Retrieval** sub-module is provided in Appendix B.2.

**Level-2 Omnimodal Dialog.** In Level-2, the primary dialog model is allowed to actively generate textual queries in arbitrary time. Specifically, we fine-tune RoboEgo to generate two groups of query words formatted as <retr>:\n<group1>\n<group2><answer>, with group1 being the "relation query" and group2 being the "keyword query". Each group is a sequence of query words separated by comma. When the final <answer> token is generated, EgoMem activates a textual retrieval process to update the *Level-2 MemChunk*. The training process is introduced in Section 4.

**Level-2 Memory Management.** The only differences to Level-1 include the **Memory Extractor** is prompted to also extract new *relation* facts from the raw dialog contents (e.g., User A says he is the boyfriend of User B now), and the **Memory Update Agent** is prompted to link the user to existing users accordingly, updating the edges of the social graph.

# 4 TRAINING APPROACH

We fine-tune RoboEgo to generate personalized response with Level-1/2 EgoMem. We also train the Episodic Trigger to label the dialog boundary for memory extraction. Interestingly, the data collection for these three tasks can be unified by different *supervision masks* on the same token stream.

## 4.1 DATA COLLECTION

**Audio Dialogs.** We collect textual transcripts simulating the lifelong personalized scenarios utilizing both Level-1 and Level-2 EgoMem. We synthesize user profiles and social graphs, collect open-sourced dialog datasets, and generate ground-truth personalized answers using large (visual-)language models. The textual transcripts are converted to audiovisual dialogs using text-to-speech (TTS) models, followed by audio augmentation to improve robustness. Details are provided in Appendix C.

**Token Stream Organization.** Multiple dialog sessions from different users are tokenized and concatenated, forming *token streams*. With a probability of 0.3, a later user instruction *interrupts* an ongoing model response, simulating the most widely-considered full-duplex scenario. The concatenated waveform are tokenized with a Mimi tokenizer (Défossez et al., 2024), formatted

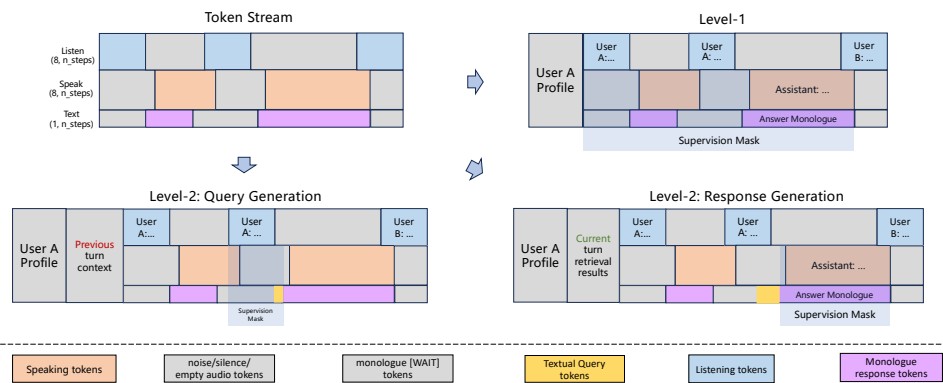

**Figure 4: Token stream structure and supervision mask for EgoMem training data**.

as described in Appendix A, and truncated to a maximum length of 8192 steps. For the textual monologue channel, we record the actual start point of each audio response, and position the textual response tokens to start 2 steps earlier. We follow the *natural* monologue strategy (Yao et al., 2025a) instead of applying word-level alignment between text and audio (Défossez et al., 2024). For each training sample with 8192 steps, positions 0-511 are reserved for the *Level-1 MemChunk*, and positions 512-768 for the *Level-2 MemChunk*, with dialogs beginning after these reserved slots. Figure 4 (top left) illustrates our token stream organization.

## 4.2 TRAINING WITH SUPERVISION MASKS

We apply three different kinds of *supervision masks* on the token streams introduced above, supporting the following three tasks:

*For Level-1 EgoMem,* for each of the $N$ multi-turn dialogs in a token stream, we position the corresponding user's profile in *Level-1 MemChunk*, and set the supervision mask to 1 for the textual and speak tokens only in the time span of the corresponding dialog, and 0 for other time steps, producing $N$ distinct training samples in total (Figure 4, top right).

*For Level-2 EgoMem,* more fine-grained supervision mask is applied to each turn $t_i, i \in [0, T_j)$ for each dialog $d_j, j \in [0, N)$ in a token stream. Specifically, for query word generation, we maintain the content of *Level-1 MemChunk* and *Level-2 MemChunk* for the previous turn, position the ground-truth query words right before the textual response for current turn, and set supervision mask to 1 from the audio start point of the current turn until the end of query words span (Figure 4, bottom left). For personalized response generation, we conduct textual retrieval with the ground-truth query words, gather the supporting fact from the connected users in the relation graph, and fill them into the *Level-2 MemChunk*. *Level-1 MemChunk* is filled with current user profile. With these contexts, we supervise on the textual and speaking tokens from the end of query words to the audio end point of dialog, including the full response utterance (Figure 4, bottom right). To summarize, for each turn, the query words and personalized response are supervised separately with different contexts, yielding $2 * \sum_{j=0}^{N} T_j$ distinct training samples from each token stream.

*For Episodic Trigger,* we leverage a full supervision mask. Each time step in the token stream is labeled following the paradigm in Section 3.1.3.

The training configurations for the above three tasks are introduced in Appendix D.

## 5 EXPERIMENTS

We focus on the following three research questions: (1) Do our retrieval sub-modules correctly identify users and recall the relevant contents? (2) Does the episodic trigger detect the correct boundaries of omnimodal dialogs? (3) Does the fine-tuned RoboEgo model effectively leverage the Level-1 and Level-2 EgoMem to deliver lifelong personalized responses? We answer these questions with quantitative results on dedicated benchmarks. As the first work in lifelong memory agents for

**Table 1:** Face verification, speaker recognition, and text retrieval results for EgoMem sub-modules.

| Tasks | Face Verification | Speaker Recognition | | | Text Retrieval |
|---|---|---|---|---|---|
| Metrics | Accuracy | pass@1 w/o s-norm | pass@1 w/ s-norm | EER | pass@5 |
| Results | 0.984 | 0.958 | 0.965 | 0.00892 | 0.960 |
| Elapsed time (s) | 0.2 | 0.1 | | | 0.1 |

full-duplex omnimodal systems, we will also release part of our test set for personalized omnimodal dialog generation to benefit future research.

## 5.1 RETRIEVAL EVALUATION

We present the benchmark and settings to evaluate the retrieval capabilities for EgoMem. The retrieval results are summarized in Table 1.

**Face Verification.** We benchmark the face retrieval module on the Labeled Faces in the Wild (LFW, (Huang et al., 2008)) dataset, achieving an accuracy of 98.4%, which is consistent with public results [1]. As the open-sourced solution demonstrates satisfying face verification performance and robustness to variations in pose and angle, we directly integrate it in our system without additional fine-tuning. The face retrieval module processes one query within 0.2 seconds with a single Nvidia H100 GPU.

**Speaker Verification.** We construct our evaluation benchmark from the public VoxCeleb (Nagrani et al., 2017) speaker verification dataset. To enable adaptive s-norm (Karam et al., 2011; Cumani et al., 2011) which is widely agreed to benefit the task, we leverage SeedTTS-eval (Anastassiou et al., 2024) as the source of imposter cohorts, using 1,000 Chinese speech embeddings as the candidate cohort set for the queries, and 2,000 embeddings for the keys. We set up a cohort number of 200 (i.e., for both the queries and the keys, the 200 closest utterances in the candidate cohorts are used to compute the mean and variance statistics for adaptive s-norm).

To assess the impact of adaptive s-norm, we first synthesize a retrieval task with 1,000 query utterances and 120 key utterances from different speakers in VoxCeleb, and compare the pass@1 with or without adaptive s-norm. We observe a moderate improvement from 95.8% to 96.5% with adaptive s-norm, confirming its benefit for retrieval stability.

Next, we sample a more challenging speaker verification test set from VoxCeleb with a highly imbalanced ratio of positive (same-speaker) to negative (different-speaker) pairs of 1:119, yielding 5,000 samples in total. Our speaker verification module achieves an Equal Error Rate (EER) of 0.89% on this benchmark with a decision threshold of 4.63. For deployment, we adjust the threshold to 6 based on human case studies to balance precision and recall. The whole speaker verification system takes less than 0.1 seconds for a retrieval run with more than 1,000 candidate entries.

**Text Retrieval.** In Level-2 EgoMem, text retrieval is used to gather relevant information w.r.t the relation and keyword queries. We therefore focus on the pass@5 metric, as it measures the ability of the system to return all relevant facts within a textual window shorter than 256 tokens (the size of the *Level-2 MemChunk*). We construct a benchmark of 200 queries sampled from our personalized dialog transcripts, with a candidate entry pool of 500 (relation, personal fact) texts. Using the straightforward retrieval strategy described in Appendix B.2, the system achieves a pass@5 score of 96%. The full system latency is controlled to be under 0.1s with a single Nvidia H100 GPU.

## 5.2 EPISODIC TRIGGER EVALUATION

We hold out a test set from the collected token streams (Section 4.1) for episodic trigger evaluation, containing 1,000 samples. We use two types of metrics: (1) *Jaccard score* measuring the overlap of dialog session spans; (2) *span_match@N* which measures precision, recall, and F1 scores for detected dialog boundaries, allowing a tolerance of $\pm N$ steps from the ground truth.

---

[1]https://github.com/serengil/deepface/tree/master/benchmarks

**Table 2:** Episodic trigger evaluation results.

| Metrics | Jaccard | P/R/F1@0 | P/R/F1@5 | P/R/F1@10 | Elapsed time (s) |
|---|---|---|---|---|---|
| Clean | 0.992 | 0.857/0.857/0.857 | 0.986/0.986/0.986 | 0.986/0.986/0.986 | 0.08 |
| Noised | 0.989 | 0.790/0.788/0.789 | 0.983/0.981/0.982 | 0.984/0.982/0.983 | |

**Table 3:** Personalized dialog evaluation results.

| Models | Level-1 | | | Level-2 | | |
|---|---|---|---|---|---|---|
| Metrics | Fact Score | Answer Quality | Throughput (fps) | Fact Score | Answer Quality | Throughput (fps) |
| Clean | 0.959 | 9.170 | 21.73 | 0.895 | 8.970 | 20.56 |
| Noised | 0.931 | 9.020 | | 0.876 | 8.820 | |

The results are presented in Table 2. At $N = 5$, our episodic trigger achieves an F1 score of more than 0.98 under both clean and noised environments, indicating robust "Valid Audio Detection" (VAD) and user session splitting capabilities within a deviation of less than $(5/12.5 = 0.4)$ seconds. Notably, noisy environments can significantly affect the prediction of more fine-grained boundaries (i.e., less than 0.2s), as we observe a large gap on *span_match@0*. This is intuitive since it takes time to figure out whether a voice indicates the start of a new user's session or just another period of noise. The episodic trigger takes 0.08 seconds to annotate a 10-minute chunk with 8192 time steps.

### 5.3 PERSONALIZED DIALOG EVALUATION

We hold out a test set from the masked token streams (Section 4.1) to assess the quality of personalized responses produced by RoboEgo, integrated with Level-1 and Level-2 EgoMem. For each dialog turn, we provide an evaluator model with the following inputs: the user instruction (textual transcript), the ground-truth textual response, the contents of the *MemChunk*s, and the textual monologue response generated by RoboEgo. The evaluator is implemented with the DeepSeek-V3 API, prompted to return two scores: (1) *Fact Score*: A binary 0/1 metric for each turn indicating whether the model's response is personalized to the user and consistent with the user profile, without factual errors. (2) *Answer Quality*: A score from 0 to 10 for each turn measuring the general helpfulness and quality of the response with respect to the user instruction, regardless of personalization.

We present the results in Table 3. We observe that for both Level-1 and Level-2 EgoMem, the models successfully achieve the expected RAG capability based on the retrieval results present in *MemChunk*s. RoboEgo achieves lower *Fact Score* in the Level-2 task, largely due to more frequent *MemChunk* updates and error cascading from the textual retrieval module. For the *Answer Quality* scores which are independent of the retrieval results, the gap becomes smaller, indicating that neither Level-1 nor Level-2 EgoMem significantly degrades the base instruct-following capability of RoboEgo.

We observe a slight drop in throughput with Level-2 memory as it introduces a longer *MemChunk*. Yet, this latency is negligible in the user experiences of full-duplex real-time chatting, as the model generates audio frames in more than 20 fps, significantly exceeding the minimum requirement for real-time audio decoder (12.5 fps).

## 6 CONCLUSION AND FUTURE CHALLENGES

In this work, we explored lifelong memory for full-duplex omnimodal models. We first defined the task and outlined the core functions, and then introduced our proposed memory system, EgoMem: Level-1 (profile-based) and Level-2 (content-driven). We integrated EgoMem to the omnimodal dialog model, RoboEgo, as an implementation example. Experimental results demonstrate that, for the first time, an omnimodal dialog agent can be equipped with robust lifelong personalization capabilities, establishing a strong baseline to support future research. Due to computational constraints, we did not explore larger model sizes or more advanced functionalities, such as complex tool use. Future directions include extending the profile/graph memory to encompass procedural memory and multimodal contents, as well as investigating whether trainable parameters can replace some of the complex agent modules and memory units.

## ETHICS STATEMENT

The data used to train the three tasks supporting our EgoMem agent is derived from synthetic transcripts generated by publicly accessible large language models. No real-world users are involved in this process, and no privacy is compromised during data collection. EgoMem is a plug-in methodology that can be applied to a wide range of models. The content generated by the dialog models does not reflect the views or opinions of the authors or affiliated institutions.

## REPRODUCIBILITY STATEMENT

We provide comprehensive details of the system paradigm, implementation, and training configurations in Sections 2, 3, 4, as well as in the Appendix. We will release a portion of our test set along with the associated code for stream organization, inference, and evaluation. The functions and signals used in Level-1 and Level-2 EgoMem are clearly defined, which we believe will support the community in reproducing our agent system and developing future variants.

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

## A    INTRODUCTION TO THE ROBOEGO MODEL

For audio signals $a_t$, we use the Mimi tokenizer [2] to extract features at 12.5 frames per second. Each audio frame is represented by one semantic token and seven acoustic tokens. The audio input and output are divided into two channels, *listening* and *speaking*, while textual monologue tokens are placed in an additional textual channel. Thus, . They are additively merged into the input embedding. The model then processes all the historical input embeddings with a 7B LLM backbone to generate the hidden state for the current time step. Following the RQ-Transformer architecture (Yang et al., 2023; Zhu et al., 2024), a lightweight *depth* Transformer (with 100M parameters) first generates a textual monologue token based on the current hidden state on the top layer, and then generates eight speaking tokens autoregressively. In lifelong deployment scenarios, this process runs continuously in real time, forming the main dialog stream, while visual signals $v_t$ are encoded through a Visual Transformer (Dosovitskiy et al., 2020) and added into the context in a time-division multiplex (TDM) manner, at fixed intervals of 2–4 seconds. We refer the readers to the related work (Yao et al., 2025b; Défossez et al., 2024; Yao et al., 2025a) for detailed structural configurations.

## B    EXTRA DETAILS FOR EGOMEM SUB-MODULES

### B.1    LEVEL-1 SUB-MODULES

**Episodic Trigger.** The episodic trigger is an RQ-Transformer-based (Défossez et al., 2024) model which shares the input stream organization and model structure topology with RoboEgo (Section 2.1), despite being much smaller with 100M parameters. It consumes the 17 audio-text channels with a maximum time step of 8192 as input. Instead of generating dialog responses, it assigns a label to each time step with the following paradigm: {0: *no dialog*; 1: *start of a new user's dialog session*; 2: *in-session step*; 3: *end of current user's session*}. We modify the attention mask from a GPT-like (Brown et al., 2020) causal mask to a Bert-like (Devlin et al., 2019) full mask, as the sequence labeling process is offline and chunk-wise. The training configurations of episodic trigger is detailed in Section 4. The evaluation results are presented in Section 5.2.

**Memory Extractor.** The memory extractor is implemented as the following pipeline:

- The episodic trigger labels each time step. According to the labels, audio chunks starting with label "1", ending with label "3", and correctly filled with "2" are considered as the audio source for one user's dialog session. The corresponding audio waveforms are clipped.

- The clipped waveform in the listening channel goes through automatic speech recognition (ASR). Raw ASR results are fed into the memory extractor. The response texts from the monologue text channel of the dialog model are also provided as reference.

- We leverage a DeepSeek-V3 (Liu et al., 2024) API to extract meaningful contents to store in the memory. Specifically, we prompt the model to summarize the dialog content with short, precise sentences, generate sentences describing the facts about the user, and figure out a 90-dimensional personal trail (Li et al., 2025a; Kang et al., 2025) for the user. The user name (or "unknown_user") is stored in a separate field.

**Memory Update Agent.** This is a DeepSeek-V3 API prompted to solve profile conflicts and formalize the extracted contents from the memory extractor, making sure that they are suitable for updating the structured memory storage.

### B.2    LEVEL-2 SUBMODULES

**Textual Retrieval.** The textual retrieval system gathers the top-K relevant textual information according to the query words generated by RoboEgo, and updates the content of *Level-2 MemChunk*. Specifically, for each user $U$ connected to current user $A$ in the social graph, $U$'s name and relation with $A$ are concatenated with each of $U$'s memory items (facts, dialog history, etc.) to form one candidate document for retrieval. If the "relation query" group is not empty, we first match all relevant

---

[2]https://huggingface.co/kyutai/mimi

users' documents via the BM25 (Robertson et al., 2009) algorithm using only the relation queries; next, if the "keyword query" group is not empty, we concatenate all the keyword into one string, and re-rank the retrieved documents based on their vector distances to this keyword string. We leverage the BGE-small (Chen et al., 2024a) model as the textual encoder. The top-K results are returned to the textual channel of *Level-2 MemChunk*.

## C  DATA COLLECTION DETAILS

**Textual Transcripts** *For Level-1 EgoMem,* we first use DeepSeek-V3 (Liu et al., 2024) to synthesize 500 user profiles including name, dialog history, and a 90-dimensional persona. Next, we synthesize 10k dialogs between user and AI assistant based on open-source instruct-following datasets (e.g., Infinity-Instruct (Zhao et al., 2024), WizardLM (Xu et al., 2023), and multimodal question answering datasets involving visual inputs (Chen et al., 2024b; X.AI, 2024)): the user instructions are retained, while responses are refined by DeepSeek-V3 (or Gemini-2.5-Pro (Gemini, 2024) for VQA) to be more helpful and personalized. Finally, we prompt DeepSeek-V3 to include more question styles in which users ask questions regarding their own dialog history and profiles. Each dialog typically contains 3–5 turns.

*For Level-2 EgoMem,* we first synthesize 500 possible relations (e.g., "father", "colleague"), construct relation graphs linking one main user with 3–5 socially connected users. We prompt DeepSeek-V3 to generate questions requiring relation-graph reasoning (e.g., "Does my mother like physical exercise?") and mix these questions with general instructions, producing 5k dialogs. For each question, the model annotates the effective query words (including both relation query and keywords query, both can be empty) sufficient to retrieve supporting facts from the profiles of connected users. After generating the query words, the model should also provide the ground-truth personalized response for training.

**TTS and Augmentation.** Audio dialogs are synthesized from the collected transcripts. Each user utterance is assigned a random human voice, and converted into speech with Fishaudio TTS (Liao et al., 2024), while model responses are consistently generated with a single fixed voice. For the listening channel, we add diverse noise from sources like DNS Challenge (Dubey et al., 2023) and RNNoise[3], as well as random speech clips. Following Moshi (Défossez et al., 2024), we also simulate microphone echo by mixing the speaking channel into the listening channel with probability 0.3, applying random gain (0-0.2x) and delay (0.1-0.5s).

## D  TRAINING DETAILS

*For Level-1 EgoMem,* we fine-tune RoboEgo with a dataset containing 158K samples with different (*Level-1 MemChunk*, token stream context, supervision mask) combinations. We duplicate the dataset by using both the original and noise-augmented listening channel. Training starts from one of RoboEgo's SFT checkpoints, running for 5 epochs with batch size 64 and a cosine learning rate decay from 1e-5 to 1.5e-6.

*For Level-2 EgoMem,* RoboEgo is fine-tuned on 54K samples with valid *Level-2 MemChunk* and the corresponding query words, combined with 50% of the Level-1 training dataset reformatted with empty *Level-2 MemChunk* and query words. As with Level-1, the dataset is duplicated with clean and noisy listening channels. Training resumes from the same RoboEgo checkpoint as in Level-1, running for 1 epoch with batch size 64 and a cosine learning rate decay from 1e-5 to 1.5e-6.

*For Episodic Trigger,* the sequence tagging model is initialized randomly. We train the model with 100K clean samples and 100K noised samples. The number of epoch is set to 45. We use a batch size of 64 and the learning rate decays from 1e-4 to 1e-6 following a cosine schedule.

---

[3]https://github.com/xiph/rnnoise

# E  REBUTTAL REVISION

This section addresses common concerns raised during the review process.

## E.1  REAL-WORLD DEMO CASE

We provide a 2-minute anonymous video (link: `https://figshare.com/s/ebd3210db6b0a47149b7`) demonstrating the performance of a deployed version of EgoMem in real-world omnimodal chatting, which showcases the generalization capabilities from synthetic training to real-world application.

## E.2  JUSTIFICATION OF LLM EVALUATION

**Correlation with Human Scores.** To assess the alignment between our LLM-based scoring and human judgment, we recruited graduate students specializing in AI to annotate model responses on a 50-turn subset of our test set. Evaluators were provided only with the initial prompts used for the model API and were instructed to assign Level-2 fact/quality scores based on the established guidelines. We computed Cohen's Kappa (McHugh, 2012) coefficient for the Fact Score as it is categorical, and the Pearson (Sedgwick, 2012) and Spearman (Myers & Sirois, 2004) coefficients for the Answer Quality scores. These metrics are standard metrics for validating evaluation methodologies. We averaged the coefficients across different annotators. The results are presented in Table 4. We observe all coefficients exceed 0.6, indicating a strong positive correlation between LLM and human evaluations.

**Table 4:** Alignment analysis between LLM and human scores.

| Fact Score | Answer Quality | |
| --- | --- | --- |
| Kappa | Pearson | Spearman |
| 0.683 | 0.621 | 0.624 |

## E.3  ROBUSTNESS IN REAL-WORLD ENVIRONMENTS

**Human Evaluation with Real-world Audiovisual Dialogs (no-memory).** While there is substantial agreement between LLM and human evaluations regarding Fact Score and Answer Quality, we acknowledge concerns regarding the robustness and user experience of a model trained on synthetic data. To address this, we conducted a comparative analysis with Qwen-2.5-omni in real-world audio dialogs, employing the same human evaluation metrics used in the backbone model (Yao et al., 2025b). We observe that EgoMem maintains a competitive advantage in key metrics related to the audio chatting experience, including Naturalness, Responsiveness, and Robustness. Notably: (i) as Qwen-2.5-omni lacks memory capabilities, we evaluated using random daily queries rather than memory-dependent ones; and (ii) the helpfulness score is significantly higher than that reported by RoboEgo, which is attributed to the differing difficulty distributions of the instructions. These results, combined with our demo video, demonstrate that training on synthesized data yields robust dialog experiences.

**Table 5:** Comparison to Qwen-2.5-omni on omnimodal dialogs in real-world environments.

| Model | Helpfulness | Naturalness | Responsiveness | Robustness |
| --- | --- | --- | --- | --- |
| Qwen-2.5-omni | **8.2** | 8.0 | 8.2 | 7.7 |
| RoboEgo+memory | 8.1 | **8.1** | **8.7** | **8.2** |

## E.4  ABLATION/CLARIFICATION ON THE SUB-MODULES' ROLES

We clarify the role of different sub-modules with ablation studies when necessary.

**Episodic Trigger vs. Rule-based Session Splitting.** We further clarify the contributions of specific sub-modules through ablation studies.

**Episodic Trigger vs. Rule-based Session Splitting.** Given the novelty of the Episodic Trigger in our architecture, we provide additional results to justify its necessity. We selected 30 recorded streams, each containing multiple dialogs with distinct user voices. We established a baseline session-splitting solution that relies primarily on the face/voice retrieval system to identify speaker changes and mark dialog boundaries, subsequently applying overlapping rules to align ASR timestamps with these boundaries.

We compared this baseline against our proposed memory extraction pipeline utilizing the Episodic Trigger. The extracted memory from each stream was subjected to a blind win-tie-lose human annotation. The results are summarized in Table 6.

**Table 6:** Ablation analysis: Episodic Trigger vs. Rule-based.

| Episodic Trigger Wins | Tie | Rule-based Wins |
|---|---|---|
| 14 | 10 | 6 |

**Impact of ASR and External LLM in Memory Extractor.** We evaluate the impact of the ASR module and the External LLM within the memory extractor using 30 Level-1 stream cases, comparing the human-annotated Fact Score on immediate factual questions about the extracted content. We test two ASR systems with different word error rates (WER), each under two conditions: (i) storing raw ASR transcripts as memory and (ii) using an External LLM to summarize and refine the content. Results are shown in Table 7.

**Table 7:** Ablation analysis: ASR and External LLM.

| ASR WER | Fact Score: Raw ASR | Fact Score: EgoMem |
|---|---|---|
| 5.9 | 0.73 | 0.87 |
| 3.0 | 0.87 | 0.9 |

We observe that when raw ASR transcripts are used directly, the ASR model's WER significantly affects the Fact Score, largely due to noisy or missing transcriptions of user instructions. In contrast, when an External LLM is applied, it jointly analyzes the user's ASR output and the dialog model's monologue—which is typically high-quality once the model correctly interprets the user speech. Leveraging this dual input, the External LLM effectively repairs imperfect ASR outputs, making the system more robust to ASR noise and variations across ASR models.

**More Clarification on the role of Face/Speaker Verification Modules.** We clarify that the results presented in Table 3 are based on the test token streams with the ground-truth users, which actually measures the model's listening and dialog generation capabilities, as well as the text retrieval quality for Level-2. If the user identification itself fails, the Fact Score will be zero. Thus, it is reasonable to directly multiply a 0.96~0.98 scale factor on the Fact Scores to measure the Fact Scores of the integrated system.

E.5    BREAKDOWN ANALYSIS OF BAD CASES

We select 50 bad cases in the test corpus of Table 3 and hand-checked the full pipeline for attribution analysis. As mentioned above, the possible failure modes include incorrect understanding of user instructs (**Class-1**), failure in recalling relative Level-2 information from textual retrieval (**Class-2**), and failure in aggregating the MemChunk information into the answer (**Class-3**). The distribution of the error types are presented in Table 8. The majority of failures comes from the listening and audio understanding capability, while the memory system itself contributes a smaller portion.

**Table 8:** Breakdown analysis of bad cases.

| Class-1 | Class-2 | Class-3 |
|---|---|---|
| 68% | 22% | 10% |

