# OpenReview forum: "EgoMem: Lifelong Memory Agent for Full-duplex Omnimodal Models"
_ICLR.cc/2026/Conference — Submitted to ICLR 2026_

### Official Review · Reviewer_XmQx · 2025-10-30

**Soundness:** 3
**Presentation:** 2
**Contribution:** 3
**Rating:** 2
**Confidence:** 3

**Summary:**

The paper presents **EgoMem**, a lifelong memory agent for **full-duplex, omnimodal** (audio-visual-text) dialogue.
The system is composed of three processes:
(1) a polling-based retrieval module that performs face and voice verification followed by multimodal content retrieval;
(2) an omnimodal dialogue backbone that integrates retrieved results via a **MemChunk** conditioning mechanism;
(3) a memory management unit that detects conversation boundaries and continuously updates long-term user memories.
The paper further defines two system levels: **Level-1**, which focuses on profile-based personalization, and **Level-2**, which introduces content-driven retrieval over a social-graph structure.
Experiments report high accuracy for sub-modules such as recognition, retrieval, and boundary detection, and achieve strong LLM-as-judge ratings for personalized dialogue quality, all within real-time latency.

**Strengths:**

1. **well designed system decomposition**
   The authors clearly separate the three main modules—retrieval, dialogue and memory management. This modularization demonstrates careful engineering and facilitates potential scalability to larger multimodal agents.

2. **practical MemChunk conditioning mechanism**
   The authors provide a concrete and lightweight solution for injecting long-term memories into ongoing dialogue without retraining or heavy memory fusion, reflecting a good understanding of real-world system constraints.

3. **meaningful distinction between Level-1 and Level-2 personalization**
   Level-1 focuses on static user profiles and personalized attributes, while Level-2 extends this to dynamic retrieval from a user’s social-graph knowledge base.  This hierarchical personalization framework is well motivated and provides a natural trade-off between performance and complexity.

4. **strong sub-module evaluations**
   Detailed results are given for the face recognition, speaker verification and retrieval modules, including EER, accuracy, and latency statistics.  Such fine-grained reporting makes the work reproducible from a system benchmarking perspective.

**Weaknesses:**

1. **Overly narrow problem setting**
   The paper explicitly limits its experiments to single-speaker full-duplex dialogues and does not handle overlapping or concurrent speech.
   This simplification makes the system much less representative of real-world applications such as in-car assistants, conference agents, or smart-home dialogue systems, where multiple speakers often interact simultaneously.

2. **Heavy reliance on synthetic and TTS-generated data**
   The training and evaluation data are largely synthesized using TTS and scripted dialogues, leading to a clear real-world gap,which weakens the realism of both acoustic and conversational characteristics, such as accent diversity, background noise, and spontaneous speech.
3. **Circularity and bias in LLM-based judging**
   The same DeepSeek-V3 is used for both generating memory content and evaluating dialogue quality, which risks introducing correlated biases.  This design choice can lead to inflated factuality or personalization scores, since the evaluation model shares the same prior with the generator.  Without human assessment or cross-model evaluation, the reported improvements lack objectivity and external credibility.

4. **Insufficient robustness analysis for the Level-2**
   Although the Level-2 model achieves strong retrieval accuracy on synthetic datasets, the paper does not analyze how recognition and retrieval errors propagate through the system.  The lack of an error-cascade analysis makes it unclear how failures in earlier modules (e.g., speaker verification or retrieval mismatch) affect dialogue correctness and user experience.  This omission undermines the claim that the system achieves stable and robust lifelong personalization.

5. **Missing ablations on critical design parameters**
   The paper does not provide ablation experiments for polling frequency, MemChunk allocation, retrieval method variants, or ASR performance under noise.  Without these studies, it remains unclear which components most affect system performance and which are sensitive to hyperparameter changes.  This absence of detailed ablation limits interpretability and reduces the explanatory power of the results.

**Questions:**

1. **Need for double-blind human evaluation**
   The paper currently relies entirely on LLM-as-judge metrics to assess dialogue quality and personalization. While automated evaluation is convenient, it often fails to capture subtle aspects such as emotional alignment, context recall, and long-horizon coherence.  Could the authors conduct or plan to conduct **double-blind human evaluations** with multiple annotators to validate the claimed improvements?

2. **Handling of overlapping and rapidly switching speakers**
   The current design appears to assume a single active speaker in each time window, which may not hold in realistic full-duplex environments.  Real-world audio streams often involve overlapping speech, interruptions, and rapid turn-taking between users.  Could the authors elaborate on how the system might be extended to handle **multi-speaker overlap or rapid speaker switching**?

3. **Ablation studies on critical design parameters**
   The paper introduces several hyperparameters and architectural decisions—such as polling interval, MemChunk allocation size, retrieval strategy, and verification thresholds—but does not analyze their influence.  These choices likely affect latency, memory usage, and personalization quality in different ways.  Would the authors provide **systematic ablation experiments** to reveal how each parameter contributes to overall system performance?

4. **Analysis of Level-2 error cascades and failure modes**
   Level-2 integrates multiple sub-modules—recognition, retrieval, and dialogue generation—each of which can introduce errors that propagate through the pipeline.  Understanding how upstream recognition or retrieval failures influence downstream dialogue responses is essential for diagnosing model stability.  Could the authors provide **quantitative error-cascade analyses or failure case studies** to clarify how the system behaves under noisy or incorrect retrieval conditions?

---

> ### Author Response · Authors · 2025-11-27
> **Thanks for your review**
>
> Thank you for your thorough review and helpful comments. We have made corresponding revisions to the paper and would like to clarify the key concerns you raised:
>
> Weakness
>
> (1) We agree that multi-speaker overlap and fully simultaneous interaction remain open challenges for omnimodal assistants. These are difficult problems not yet solved by existing systems. Nevertheless, we would like to clarify that our model **can handle rapid speaker switching** down to the minimum interval determined by our polling frequency (2 seconds). This is because our training data includes scenarios where a user interrupts 1 second after the model starts responding. In practice, switching intervals shorter than 2 seconds are uncommon in real-world usage, and many overlapping utterances do not imply a user switch.
>
> Regarding **simultaneous multi-user interactions**, we acknowledge that our current system cannot provide personalized responses to multiple users visible on the screen at the same moment. A possible extension is to maintain multiple face-detection tracks combined with lip-movement detection—an approach fully compatible with our current architecture. Ultimately, accurately identifying the active speaker in such scenarios is a classic **cocktail party problem**, which is beyond the scope of this paper, as we also note in the manuscript.
>
> (2),(3):  For real-world performance concerns, we have added new materials and revisions—please see our response to all reviewers. For concerns regarding LLM-based evaluation, we additionally provide **human–LLM correlation scores (Appendix E.2)**, which may address the reliability questions you raised.
> Regarding the circularity issue: for the factual-consistency metric, the judge LLM must provide an explicit justification for its decision. We manually inspected these justifications and found that **>90%** were reasonable, aligning well with the high measured correlation between human and LLM ratings.
>
> (4) (5):
> We have added ablation studies, bad cases, and detailed error analyses in the revised version, which we hope address most of your concerns.
> As for the polling frequency: the 2-second interval is chosen based on the observation that most single-user utterances last longer than 2 seconds. Since polling introduces **near-zero inference overhead** (due to asynchronous processing), there is no compelling efficiency reason to increase the interval.
>
> Questions
>
> Please refer to our responses above and the newly added content in Appendix E.
>
> We hope these clarifications address your concerns. We appreciate your thoughtful feedback, which has been very helpful in improving the paper, and we remain open to further discussion.

---

### Official Review · Reviewer_X4Wc · 2025-10-30

**Soundness:** 2
**Presentation:** 3
**Contribution:** 3
**Rating:** 4
**Confidence:** 3

**Summary:**

The paper proposed EgoMem, the first lifelong memory agent tailored for real-time omnimodal full-duplex models. The proposed model tried to solve the humanoid cognition problem of robots deployed for a user group by exploring lifelong memory capability. EgoMem consists of three processes: the retrieval process to identify the user and gather related documents, the dialog process to realize real-time response, and the memory management process to detect dialog boundaries, extract information, and update memories. The paper follows a two-level development of chatbots to solve the user-specific service and cross-user references step-by-step. Deployed on RoboEgo chatbot, the system could handle the multi-user conversation problem with only raw audiovisual inputs, and realized high fact-consistency and answering quality scores under LLM scoring evaluation.

**Strengths:**

1. The problem is well-motivated and significant. This paper tackles the messy reality of continuous, multimodal, multi-user interaction head-on, which is a crucial and challenging direction for the field.
2. The modeling of multi-user interaction is reasonable and sound, with some special cases such as interruption considered.
3. The paper provides a detailed analysis of the data organization in training, which is well aligned with the proposed multi-user interacting model and real-world scenarios.

**Weaknesses:**

1. The proposed method is an assembly of several existing models, and the paper did not discuss the challenges this work addresses. The technical innovation of this paper appears limited, and further clarification of the novel aspects would strengthen the work.
2. As a study containing interaction with user groups in reality, this paper lacks a real user study. The experiments are entirely based on synthetic data generated by LLMs, causing concerns about the model's real-world performance.
3. The evaluation of the experiments is entirely based on LLM scoring. The LLM-based evaluation result alone is not sufficiently trustworthy. A crowdsourced study or case studies could be introduced to enhance the credibility.
4. The scores in Table 3 lack comparison to other models or human behaviors. The authors claim this is the "first try" in this field, which is used to justify the lack of comparative baselines. However, this is not entirely satisfying. Could a simpler (but still strong) baseline have been implemented? For example, a system that simply runs ASR on the stream, chunks it by speaker, and feeds it all into a large-context RAG system.
5. While the proposed EgoMeM uses the DeepSeek API extensively, it does not describe the prompts used. Including this information would improve reproducibility and clarity.

**Questions:**

1. How is this framework's compatibility with other progresses in long-term memory construction and retrieval to enhance performance?
2. Could the model update the relevant information to other users according to the conversation with the current user? For example, if user A mentions a schedule with user B during a conversation with the model, could the model update user B's profile too after this conversation?

---

> ### Author Response · Authors · 2025-11-27
> **Thanks for your review**
>
> Thanks for your insightful and helpful comments. We have made modifications to the paper (Appendix E) accordingly and would like to make the following clarifications regarding your main concerns:
>
> Weakness
>
> (1) The key challenges of current memory-agent systems are summarized in the introduction (lines 71–85), including difficulties in handling lifelong audiovisual streams, implicit dialogue boundaries, and multi-user relations. While future work may achieve these goals using a single end-to-end model, demonstrating a practical system-level solution is more feasible at the current stage.
> Indeed, we adopt existing implementations with sufficiently strong performance for certain sub-modules when appropriate. Our goal is to show that straightforward choices already work well within our paradigm, thereby demonstrating its extensibility, rather than proposing new designs for each component. This approach is also common in prior memory-agent research.
> Importantly, most modules in our system have their necessity. For example, the new ablation studies (Appendix E.4) show that our learned episodic trigger substantially outperforms rule-based dialogue-boundary detection—highlighting a novel and effective design choice for current memory systems.
>
> (2), (3): we have added modifications: please check our response to all reviewers and Appendix E.
>
> (4): Regarding comparisons to simpler baseline systems: we have added experiments in Appendix E.4, in which some ablated system can be viewed as a simpler baseline, including the **large-context RAG over raw ASR transcripts** you mentioned. Please refer to the results and conclusions in the section. Additionally, user-specific storage and retrieval is generally more efficient because the system does not need to handle non-target-user information in every query.
>
> (5): We have provided example prompts in the attachment in the page. This component is intentionally flexible—prompts can be tuned for specific applications or replaced by other modern long-term-memory (LTM) mechanisms.
>
> Questions
>
> (1) Other long-term memory systems can indeed replace parts of our memory management and textual retrieval modules. However, our framework remains unique in its omnimodal capabilities and native full-duplex interaction, which are not covered by existing approaches.  Please also check our response (2) to reviewer 5v1E.
>
> (2) As noted above, this functionality can be easily extended by modifying the prompts used by the memory-management module.
>
> We hope these clarifications address your concerns. We appreciate your thoughtful feedback and are open to further discussion. Thank you again for your constructive comments.

---

### Official Review · Reviewer_rerw · 2025-10-30

**Soundness:** 2
**Presentation:** 2
**Contribution:** 3
**Rating:** 6
**Confidence:** 4

**Summary:**

The paper introduces EgoMem, the first lifelong memory agent designed for full-duplex omnimodal models that process real-time audiovisual streams. The agent enables real-time personalized dialogue and interaction with users by maintaining long-term memory of users' preferences, social relationships, and facts derived from audiovisual history. The EgoMem system operates asynchronously through three processes: retrieval (for recognizing users and gathering context), real-time, full-duplex, omnimodal dialog generation, and memory management. It is evaluated in real-time personalized dialogue scenarios using RoboEgo, an omnimodal chatbot, and shows promising results with high accuracy and fact-consistency scores.

**Strengths:**

1. This study addresses an important problem, introducing a comprehensive and useful pipeline, which integrates lifelong memory into real-time full-duplex omnimodal systems.
2. The paper presents a well-organized and structured approach to EgoMem's operation, including its division into Level-1 (profile-only) and Level-2 (content-driven) memory. The description of each module (retrieval, dialog, and memory management) is clear, and the use of both face and voice recognition, along with text retrieval, is well-motivated and effectively integrated.
3. I appreciate that the authors have done a substantial amount of engineering and evaluation. They evaluated each component of their complex system individually with strong quantitative results.
4. The paper demonstrates strong experimental results, especially with the retrieval and memory management modules.
5. The use of Memory in the omni-modal, real-time interaction is well-motivated and clearly demonstrated.

**Weaknesses:**

1. While I acknowledge the significant engineering effort behind the system; however, its architecture appears to rely heavily on existing off-the-shelf components (face verification, speaker verification and external LLMs for memory extraction and updating). I think I am not in the best position to assess the technical novelty of the work, and I defer that judgment to AC.

2. The entire system is trained and evaluated on synthetic data only. User profiles and connections are synthesized, dialogs are generated by LLMs, and audio is generated via TTS—without any real-world user interactions or human-subject studies, raising concerns about its generalization and effectiveness in real-world scenarios.

3. Also, the primary evaluation is performed by LLM only, there is no human evaluation to validate these results. I am skeptical that an AI judge can reliably score nuanced aspects like "helpfulness" in a personalized, long-term context.

4. There is no detailed error analysis regarding the system's failure modes or edge cases. Understanding these failure modes would offer deeper insights into its real-world applicability.

**Questions:**

1. Is user privacy taken into consideration in this model? When a query from user A contains some private information of user B, how could this model respond?

2. The results show that the Level-2 system (with the social graph) has a lower Fact Score (87.6% on noised) than the Level-1 system (93.1%). The paper attributes this to "more frequent MemChunk updates and error cascading". Could you elaborate on this? Is the query-generation model itself prone to error, or is the text retrieval  the main source of failure? Does this drop in performance suggests the L2 system is less reliable than L1 under noise?

3. As there have been continuing advances in long-term memory, how compatible is the proposed EgoMem framework with these developments?

---

> ### Author Response · Authors · 2025-11-27
> **Thanks for your review**
>
> Thank you for your careful reading and insightful comments. We are happy to provide further clarifications and discussion regarding your concerns:
>
> Weakness
>
> (1) Indeed, we use existing implementations with satisfying performance for some sub-modules when necessary. Our primary contribution lies in making the whole framework work for omnimodal, full-duplex setting. For the submodules, our goal is to show that **straightforward and lightweight choices already work well**, highlighting the paradigm’s extensibility rather than proposing new architectures for every component. This is also common practice for most related memory agent researches. This practice is also common in prior memory-agent research. That said, beyond the overall framework, we do introduce several novel technical elements, including our training algorithm for the memory-augmented full-duplex model with its three integrated tasks.
>
> (2), (3), (4): we have added modifications and supportive materials in Appendix E; please refer to our response to all reviewers.
>
> Questions
>
> (1) User privacy can be solved with minor modifications to our system: we prompt the Memory Extractor to detect and mark potential privacy information, which can then be filtered via simple rule-based blocking before textual retrieval. The model has already been trained to respond with “I do not know” when the relevant fact is not retrieved, ensuring safe fallback behavior.
>
> (2) Apologies for the ambiguity. "More frequent MemChunk updates" refers to a higher frequency of relation-related queries that trigger Level-2 textual retrieval, which increases the chance for error propagation. Given that the textual retrieval module has a pass@5=0.96, other failure cases mainly originate from failure in generating the full query keyword set. Importantly, this does not imply that L2 is less robust under noise, as shown in Table 3, where the performance degradation under noise is similar for both L1 and L2. The gap mainly arises from the longer error chain and inherently more complex reasoning required at L2.
>
> (3) Long-term memory agents can indeed serve as compatible alternatives for some components of our memory-management and textual-retrieval sub-modules. However, our overall framework remains unique in its omnimodal and full-duplex capabilities, which are not covered by existing systems. Please also check our response (2) to reviewer 5v1E.
>
> We hope the above clarifications address your concerns. We appreciate your valuable feedback and are open to further discussion. Thank you again for your thoughtful comments.

---

### Official Review · Reviewer_5v1E · 2025-11-03

**Soundness:** 3
**Presentation:** 3
**Contribution:** 3
**Rating:** 4
**Confidence:** 3

**Summary:**

This paper proposes EgoMem, a lifelong memory agent designed for full-duplex omnimodal models that process real-time audiovisual inputs. Unlike existing textual memory agents for large language models (LLMs), EgoMem directly operates on raw audiovisual streams, supporting user identification, memory retrieval, and long-term knowledge updating. The framework comprises three asynchronous processes: (1) Retrieval, identifying users and fetching related content; (2) Omnimodal Dialog, generating personalized responses in real time; and (3) Memory Management, detecting dialog boundaries and updating user-specific memory. EgoMem is integrated into the RoboEgo model and evaluated on multimodal benchmarks, achieving high accuracy for retrieval and memory tasks, and strong personalization performance in real-time dialogs.

**Strengths:**

1. Attempt to design a lifelong memory system for full-duplex omnimodal models, addressing an emerging research frontier.
2. Well-motivated framework with clear modular design (retrieval, dialog, memory management).
3. Strong experimental performance and latency benchmarks on a large-scale synthetic multimodal dataset.
4. The paper defines a new benchmark and data pipeline for personalized omnimodal dialogs, potentially impactful for future research.

**Weaknesses:**

1. Experiments rely entirely on synthetic data, limiting confidence in real-world generalization.
2. Lack of comparative baselines against other multimodal or memory-augmented agents, only presenting the results of the proposed method is not convincing.
3. Lack of results from ablation experiments on sub-modules, such as the impact of the accuracy of speech recognition (ASR) results on system final performance.

**Questions:**

1. How well does EgoMem perform with real user data, given all training is synthetic?
2. How does EgoMem compare to recent LLM memory frameworks in textual or multimodal contexts?

---

> ### Author Response · Authors · 2025-11-27
> **Thanks for your review.**
>
> Thanks for your constructive comments. We would like to respond to your concerns raised in the weakness and questions as follows:
>
> (1) **Synthetic vs real**: Please refer to our response to all reviewers for a 2-minute real-world demonstration video. This scenario is out-of-distribution relative to our synthetic training data, yet the system performs robustly, supporting the effectiveness of our approach.
>
> (2) **Comparison with other agent systems**: we would like to clarify that most current memory agents can be viable alternatives to part of our memory management prompts and the textual retrieval sub-module (some multimodal agents such as MIRIX may cover face retrieval as well with complex engineering/fine-tuning). However, comparing against these systems would mainly reflect how our prompts or BM25+BPE retrieval stack up against more complex engineering solutions. For the core purpose of this paper, it is more important to demonstrate that our overall paradigm functions effectively even with minimal and basic component implementations. Practically, we are currently the only agent system that provides the full pipeline in full-duplex omnimodal settings, and thus no direct alternative solution currently exists.
>
> (3) We have updated **ablation studies** in Appendix E, including the impact of ASR quality on system performance.
>
> We hope these clarifications address your concerns. We appreciate your feedback and welcome further discussion. Thank you again for your valuable comments.

---

### Author Response · Authors · 2025-11-27
**Response To all Reviewers**

To all reviewers:
We sincerely thank the reviewers for their thoughtful and constructive feedback. We believe the comments have significantly helped us improve the quality and clarity of the paper. Below we summarize the key additions and clarifications in our revised manuscript. Most of these updates appear in the new **Appendix E**.
- **Response to Common Concerns**
  - **Real-world case**: Since several reviewers asked about real-world performance, we now provide a 2-minute anonymized demonstration video (recorded in Chinese with English transcriptions). Please check: https://figshare.com/s/ebd3210db6b0a47149b7
  - **Human evaluation**: We have added a human–LLM alignment analysis in Appendix E.2, which supports the validity of using LLM-based metrics. Additionally, Appendix E.3 now includes human evaluation results on real-world dialogue interactions. These results collectively show that, although our system is trained purely on synthetic data and evaluated using LLMs, it generalizes well to real-world scenarios and aligns with human judgments.
  - **Ablation study**: Appendix E.4 provides a detailed ablation study to clarify the contributions of our sub-modules. These results can also be viewed as comparisons against plausible alternative baselines. The findings validate our episodic trigger mechanism and the structured per-user memory design.
  - Some system prompts are provided in the Supplementary Material zip, translated into English.

---

### Meta-Review · Area_Chair_fY8X · 2026-01-02

**Summary:**

The paper proposes EgoMem, a lifelong memory agent designed for full-duplex omnimodal models. The system utilizes three asynchronous processes (retrieval, dialog, and memory management) to handle real-time audiovisual streams, enabling user recognition and personalized responses based on long-term memory

**Reviewer Concerns:**

- The reviewers generally appreciated the comprehensive engineering effort and the clear modular design of the system (separating retrieval, dialog, and memory management). The distinction between profile-based and graph-based personalization was noted as a logical and well-motivated structure.

- The problem space is recognized as highly relevant and significant for the advancement of embodied AI and humanoid robotics.

- The authors made a significant effort during the rebuttal phase, adding a real-world demonstration video, conducting ablation studies on the episodic trigger, and providing an analysis of human-LLM evaluation alignment in the new Appendix E.

**Reviewer Scores:**

- A primary concern shared by multiple reviewers (5v1E, rerw, X4Wc, XmQx) is the exclusive reliance on synthetic data for training and evaluation. While the authors provided a short demo video during the rebuttal to show "real-world" capability, this seems anecdotal. For a paper claiming applicability to "lifelong, real-time, and embodied scenarios," the lack of quantitative evaluation on real-world audiovisual data (with natural noise, overlapping speech, and acoustic diversity) is a critical gap.

- The evaluation relies heavily on LLM-as-a-Judge using DeepSeek, which creates potential circularity given that DeepSeek APIs are also used within the system for memory extraction and updating. While the authors added a correlation study (Appendix E.2) showing alignment with human judgment, the sample size (50 turns) is relatively small. Reviewers X4Wc and rerw pointed out that for a dialog system, particularly one focusing on helpfulness and personalization, a robust human-subject study or a larger-scale independent evaluation is necessary.

- The contribution is viewed primarily as an engineering integration of off-the-shelf components rather than a fundamental algorithmic innovation. While system papers are valuable, the lack of comparison against stronger or more diverse baselines makes it difficult to place the magnitude of the contribution in the context of the current SOTA.

---

### Decision · Program_Chairs · 2026-01-26

Reject